# Aspartate aminotransferase-to-platelet ratio index (APRI): A potential marker for diagnosis in patients at risk of severe malaria caused by *Plasmodium vivax*

**Karla Sena Guedes**[1]☺, **Bruno Antônio Marinho Sanchez**[1]‡, **Luciano Teixeira Gomes**[2]‡, **Cor Jesus Fernandes Fontes**[1,2]☺*

1 Institute of Health Sciences, Federal University of Mato Grosso, Sinop, Brazil, 2 Júlio Müller University Hospital, Federal University of Mato Grosso, Cuiabá, Brazil

☺ These authors contributed equally to this work.
‡ These authors also contributed equally to this work.
* corfontes@gmail.com

**Data Availability Statement:** All relevant data are within the paper and its Supporting Information files.

## Abstract

Acute infection with *Plasmodium vivax*, classically associated with benign disease, has been presenting as serious and even fatal disease in recent years. Severe disease is mainly due to biochemical and hematological alterations during the acute phase of infection. In the present cross-sectional study, the aspartate aminotransferase-to-platelet ratio index (APRI) was evaluated as a method for identifying patients at risk of severe vivax malaria. This retrospective study included 130 patients with confirmed *P. vivax* infection between June 2006 and January 2018. Clinical-epidemiological data were obtained from medical records. Hematological and biochemical parameters were determined using automated equipment. The criteria of severity for infection by *Plasmodium falciparum*, established by the World Health Organization (WHO), were adapted to classify patients with danger signs of severe vivax malaria. Of the 130 patient's records evaluated, 19 (14.6%) had one or more signs and symptoms of severe malaria. The mean APRI values among patients with and without severe malaria were 2.11 and 1.09, respectively (p = 0.044). Among those with severe disease, the proportion with an APRI value above 1.50 was 30% compared to the 10% among those without severe disease (p = 0.007). The area under the receiver operating characteristic curve (95% CI), calculated to assess the accuracy of the APRI in discriminating between patients with and without severe disease, was 0.645 (0.494; 0.795). An APRI cutoff of 0.74 resulted in sensitivity of 74.0%, specificity of 56.0%, and accuracy of 65.0%. This study shows that the APRI is elevated in patients with evidence of infection by *P. vivax*. Based on the good sensitivity found in this study, we conclude that this simple index can serve as a diagnostic biomarker to identify patients at risk of severe disease during the acute phase of *P. vivax* infection.

**Funding:** Financial support provided by the Brazilian National Council for Scientific and Technological Development and State of Mato Grosso Research Foundation–PRONEX/CNPq/FAPEMAT (Malaria Network), grant number 555652/2009-2. KSG was recipient of a FAPEMAT Doctor Scholarship, Grant number 041/2016. The funders had no role in study design, data collection and analysis, decision to publish, or preparation of the manuscript.

**Competing interests:** The authors have declared that no competing interests exist.

## Introduction

Since the early 2000s, cases of severe malaria associated with *Plasmodium vivax* have been described in Brazil and other endemic regions [1–4]. In 2015, severe vivax malaria was responsible for causing around 16% of all malaria mortality outside the sub-Saharan region [5]. In Indonesia and in non-endemic areas of India, *P. vivax* infection rates have been reported to be comparable to the rates of *P. falciparum* infection [6, 7]. In Latin America, severe vivax malaria began to acquire importance in 2009, when the number of serious cases of the disease increased significantly [8–10].

The main complications associated with severe vivax malaria are thrombocytopenia, hemoglobinuria, and respiratory failure. In a systematic review, Rahimi et al. (2014) showed that patients with severe *P. vivax* may complicate with liver dysfunction, circulatory shock, respiratory difficulty, renal failure, severe anemia, and central nervous system involvement [11]. A cohort study performed in southwest India demonstrated changes in respiratory frequency and in serum levels of total bilirubin, creatinine serum, and hemoglobin dosage associated with *P. vivax* infection. These findings are now considered independent predictors for severe vivax malaria [12]. Additionally, a significant reduction in the number of platelets, leukocytes, lymphocytes, eosinophils, red blood cells, and hemoglobin has been observed in the hematologic profile of patients with acute *P. vivax* infection [13, 14].

Several serum markers have been used as tools to assess the potential severity of various diseases, including dengue, cirrhosis and viral hepatitis, and malaria [15–20]. In severe liver diseases, biochemical assays have been tested for their diagnostic and prognostic potential. One minimally invasive biochemical assay is the aspartate aminotransferase-to-platelet ratio index (APRI), determined by the patient's aspartate aminotransferase (AST) level (corrected for that enzyme's upper limit of normality in the blood) divided by the patient's platelet count [21]. A meta-analysis of patients with hepatocellular carcinoma confirmed that the APRI is associated to liver function deterioration and could be used as a stable and reliable marker to predict tumor progression. In addition, APRI values are associated with the prognosis of several other neoplasms and human infections, including chronic hepatitis C [22].

For non-hepatic infectious diseases, there is not much information using the APRI as a marker of disease severity. It is known that the liver is an important target in *P. falciparum* and *P. vivax* infections: hepatic damage was clinically demonstrated due to an intense inflammatory response and oxidative stress. Plasmodium-induced histopathological changes in the liver include hepatocyte necrosis, granulomatous lesions, hyperplasia of Kupffer cells, phagocytosis of malarial pigment, cholestasis, and infiltration of monocytes. Such changes can alter the levels of transaminases, even in cases of non-severe disease, which may return to normal after treatment [23].

Early diagnosis of malaria parasite-induced complications and their precise treatment are important for preventing disease progression and death. Thus, easily obtained biochemical markers, such as the APRI, may function as important tools for identifying the presence of hepatic damage and, consequently, identifying patients with a higher potential for disease progression during the acute phase of *P. vivax* malaria. The main objective of this study was to evaluate the APRI as a diagnostic tool for identifying patients at risk of *P. vivax* malaria severity. For this purpose, a clinical assessment (based on the severity criteria for *P. falciparum* malaria) evaluated patients for signs, symptoms, and laboratory changes indicative of severe vivax malaria.

## Patients and methods

### Study design and population

In this cross-sectional study, a retrospective analysis was performed on the routine laboratory test results of 130 patients with acute *P. vivax* infection that received care in the period from

2011 to 2018 at the Júlio Müller University Hospital in Cuiabá, Mato Grosso, Central Region of Brazil. All the study participants were outpatients and only attended the hospital for the first medical evaluation, not returning for clinical follow-up. Therefore, no follow-up information on disease progression was obtained in the study. At this hospital all suspected malaria patients undergo careful clinical and laboratory examination by doctors. Thick smear for *Plasmodium* screening, blood cell count, and basic blood biochemistry were performed at the first patient consultation. After careful review of the medical records, patients with other concomitant infections, such as hepatitis B, hepatitis C, dengue, HIV, or bacterial infection, were excluded from this analysis. Patients with arterial hypertension, ischemic cardiopathy, a history of myocardial infarction or cerebrovascular accident, diabetes mellitus, chronic liver disease, history of malignant disease, or neurological or psychiatric diseases were also excluded. Treatment for all patients was provided in compliance with the recommendations of the Ministry of Health for the treatment of *P. vivax* infections [24].

*P. vivax* infection was diagnosed by microscopic examination of the blood and subsequently confirmed by PCR in accordance with the protocol described by Snounou et al. (1993) [25]. Microscopy was used to quantify parasite density by enumerating all the parasite's asexual blood stages in 500 leukocytes.

## Classification of groups according to severity

The WHO-established criteria for defining severe *P. falciparum* malaria were adapted in order to stratify patients into groups based on disease severity in the acute phase of *P. vivax* malaria [5]. As the AST, a biomarker of liver disease severity, was used in this study, jaundice, hyperbilirubinemia, and elevated hepatic enzymes [26] were not included as criteria for severe malaria in the present analysis. Similarly, the platelet count has not been considered an associated parameter with severe malaria [27].

Adaptations were made to the criteria of acute renal insufficiency and hyperparasitemia, as these are the most questioned in the medical literature [30, 31]. The threshold level of serum creatinine was adapted to be above 1.5 mg/dL (already used in other studies [32, 31]) and the threshold for parasitic density was adapted to be over 20,000/μL, although there are references to even smaller cutoff points for parasitemia associated with severe *P. vivax* in the medical literature [33, 34].

## Determination of APRI values

The clinical and biochemical alterations used for evaluation of hepatic damage in study patients were jaundice as well as elevated levels of total bilirubin serum, alanine aminotransferase (ALT), aspartate aminotransferase (AST), and alkaline phosphatase [23]. Patient serum was analyzed by the photometric method in a BT-3000 Plus automated biochemical analyzer (Diamond Diagnostics, Cambridge, MA, USA). Patient platelet count was enumerated using a hematologic multiparameter analyzer (Pentra 80, Horiba Medical, Montpellier, France). The APRI was calculated using the following formula:

$$\text{APRI} = \frac{\frac{AST\ (UI/L)}{Upper\ limit\ of\ AST}}{\text{Plaques x } 10^9/\text{L}} \text{ x } 100$$

## Data analysis

All statistical analyses were performed using the *Stata* package version 12.0. Patient age was described and analyzed as a continuous variable. Patient age was stratified into four age groups in accordance with WHO guidelines [35]. In this study, platelet count, to identify

thrombocytopenia, was stratified into severe ($< 50{,}000$ platelets/μL) and moderate (between 50,000 and 150,000 platelets/μL), as previously described [15, 36]. Chi-squared or Fisher's exact test was used to compare categorical variables. Chi-squared test for linear trend was used to analyse trend in proportions. The Shapiro-Wilk test was used to verify whether the biochemical and hematological results had a normal distribution. Since all the parameters analyzed showed abnormal distributions, the non-parametric Mann-Whitney test was applied to compare the continuous variables between groups. Values of $p < 0.05$ were considered significant for all the analyses.

To assess the discriminatory power of the APRI as a readout for groups with and without signs of severe *P. vivax* infection, a ROC (receiver operating characteristic) curve was graphed. The area under the ROC curve was analyzed and its respective 95% confidence interval was calculated assuming a non-parametric distribution. The concordance probability as described by Liu was utilized to define the cutoff point for the APRI that yielded the greatest sensitivity and specificity values along with the lowest probability of chance occurrence [37]. This method of obtaining a cutoff point is recommended when the analysis objective is to determine values that maximize sensitivity or specificity for clinical decision making [38].

### Ethics statement

The study was conducted in accordance with the Declaration of Helsinki and approved by the Research Ethics Committee of the Júlio Müller University Hospital (CEP number 130/HUJM/ 2011). Written informed consent was obtained from all patients after suitable explanation about the research. For participants under 18 years of age, consent was obtained from parents or guardians.

## Results

We analyzed 189 medical records from eligible patients with symptomatic malaria caused by microscopy and PCR-confirmed monoinfection with *P. vivax*. Fifty-nine patients were not included in the study because lack of clinical or laboratorial data, and presence of comorbidities. The remained 130 patient's records were included in this analysis. Of these patients, 100 (76.9%) were male and 30 (23.1%) were female. Their ages ranged from 1 to 78 years with a mean of 38.4 years and standard deviation (SD) of 15.4 years. A majority (70%) of patients reported less than 7 days of symptoms (Table 1). All patients declared to have had fever in the 24 hours prior to entering the study.

Of the 130 patients, 19 (14.6%) had one or more of the signs and symptoms used in this study to indicate severe vivax malaria. The following clinical and laboratory abnormalities suggestive of hepatocyte damage were observed: jaundice (9.2%), total hyperbilirubinemia (16.2%), high alkaline phosphatase (20.1%), high ALT (14.6%), and high AST (3.9%). However, none of these clinical and laboratory abnormalities were significantly associated with indicators of severity of acute vivax malaria. However, most patients (78.5%) had low platelet counts (less than 150,000/ platelets/μL). The mean number of platelets in the severe group was 80,263 cells/μL (SD 46,621), which was significantly lower than that of patients without signs of severe disease ($p = 0.019$). Severe thrombocytopenia, between 15,000 and 100,000 platelets/ μL, was found in 17.1% of the patients. A negative association ($p = 0.032$) between degree of thrombocytopenia and presence of signs of severe disease was observed as compared to patients with normal platelet counts (Table 1).

The average APRI value was 1.24 (SD 1.49). The average APRI was significantly higher [2.11 (SD 2.64) vs. 1.09 (SD 1.14); $p = 0.044$] in the group with signs of severe vivax malaria.

**Table 1. Characteristics of the *Plasmodium vivax* infected patients, according to presence and absence of signs of severe vivax malaria.**

| Characteristic | | Sign of severity | | Total (%) | p |
|---|---|---|---|---|---|
| | | Present (n = 19) n (%) | Absence (n = 111) n (%) | | |
| Gender | *Male* | 17 (17,0) | 83 (83,0) | 100 (76,9) | 0,262* |
| | *Female* | 2 (6,7) | 28 (93,3) | 30 (23,1) | |
| Age (years) | *1–4* | 0 (0,0) | 1 (100,0) | 1 (0,8) | 0,465** |
| | *5–19* | 1 (7,7) | 12 (92,3) | 13 (10,0) | |
| | *20–49* | 10 (11,9) | 74 (88,1) | 84 (64,6) | |
| | *≥ 50* | 8 (25,0) | 24 (75,0) | 32 (24,6) | |
| | *Mean (SD)* | **43,0 (17,6)** | **37,6 (15,0)** | **38,4 (15,4)** | **0,173#** |
| Symptoms onset time (days) | *1–2* | 5 (26,3) | 17 (73,7) | 22 (16,9) | 0,431** |
| | *3–7* | 8 (11,6) | 61 (88,4) | 69 (53,1) | |
| | *≥8* | 6 (15,4) | 33 (84,6) | 39 (30,0) | |
| Jaundice | *Present* | 4 (33,3) | 8 (66,7) | 12 (9,2) | 0,054 |
| | *Absence* | 15 (12,7) | 103 (87,3) | 118 (90,8) | |
| Total bilirubin | *Normal* | 15 (13,8) | 94 (86,2) | 109 (83,8) | 0,530 |
| | *Altered* | 4 (19,0) | 17 (81,0) | 21 (16,2) | |
| ALT (IU/L) | *Normal* | 18 (16,2) | 93 (83,8) | 111 (85,4) | 0,377* |
| | *Altered* | 1 (5,3) | 18 (94,7) | 19 (14,6) | |
| AST (IU/L) | *Normal* | 18 (14,4) | 107 (85,6) | 125 (96,2) | 0,999 |
| | *Altered* | 1 (20,0) | 4 (80,0) | 5 (3,8) | |
| Alkaline phosphatase (IU/L) | *Normal* | 14 (13,6) | 89 (86,4) | 103 (79,2) | 0,519 |
| | *Altered* | 5 (18,5) | 22 (81,5) | 27 (20,8) | |
| Platelets (cells/ μL) | *15.000–50.000* | 5 (22,7) | 17 (77,3) | 22 (17,1) | 0,032** |
| | *50.000–150.000* | 13 (16,5) | 66 (83,5) | 79 (61,2) | |
| | *>150.000* | 1 (3,6) | 27 (96,4) | 28 (21,7) | |
| | *Mean (SD)* | **80.263 (47.621)** | **110.913 (57.264)** | **106.433 (56.840)** | **0,019#** |
| APRI | *0–1,50* | 10 (10,0) | 90 (90,0) | 100 (76,9) | 0,007 |
| | *> 1,50* | 9 (30,0) | 21 (70,0) | 30 (23,1) | |
| | *Mean (SD)* | **2,11 (2,64)** | **1,09 (1,14)** | **1,24 (1,49)** | **0,044#** |

AST: aspartate aminotransferase; ALT: alanine aminotransferase; IU/L: International Units per Liter; APRI: Aspartate aminotransferase-to-platelet ratio index; SD: Standard Deviation. p value determined by chi-squared test

* Fisher's exact test

** Chi-squared test for linear trend

# Man-Whitney test

APRI values above 1.50 showed a significant association (p = 0.007) with signs of severe disease during acute *P. vivax* infection (Table 1 and Fig 1).

For all APRI values, the area under the ROC curve (95% CI) was 0.645 (0.494; 0.795). A cut-off of 0.74 resulted in sensitivity of 74.0%, specificity of 56.0%, and accuracy of 65.0% for diagnosing patients at risk of severe disease due to *P. vivax* infection (Fig 2).

## Discussion

In the present study, low platelet count and high APRI were significantly associated with the presence of clinical and laboratory manifestations of severe malaria in patients during the acute phase of *P. vivax* infection. An APRI cutoff of 0.74 was reasonably sensitive for

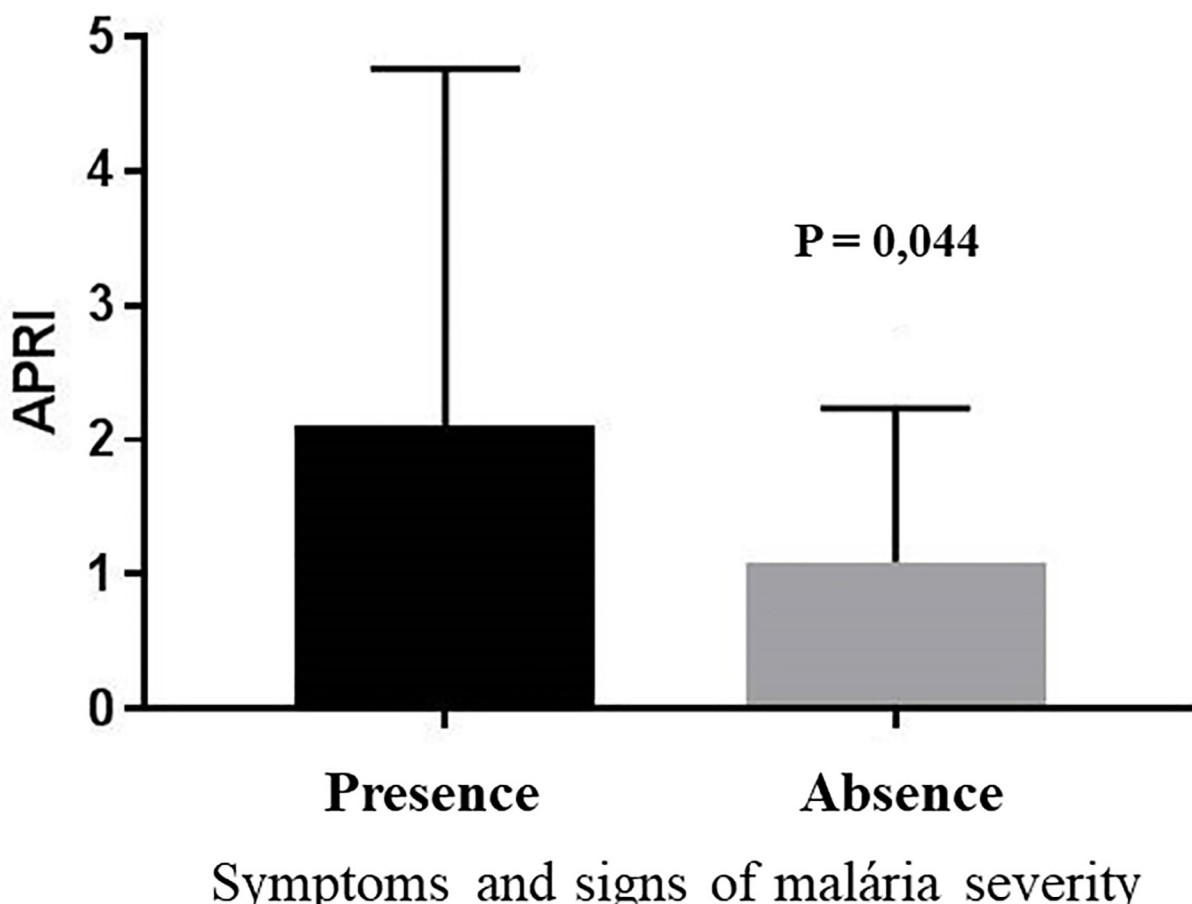

**Fig 1. Aspartate aminotransferase-to-platelet ratio index (APRI) in patients infected with *P. vivax* with presence or absence of signs and symptoms of severe malaria.**

identification of patients with a higher potential for severe disease. Despite the low specificity and the small number of patients in this sample, the discovery of the APRI as a sensitive biomarker of severe vivax malaria is very promising.

According to the WHO, studies conducted in reference hospitals in which it is possible to identify the *Plasmodium* species causing malaria provide more reliable estimates of severe malaria occurrence and mortality risk [39]. The development of simple and sensitive tools to predict severe disease is greatly needed. The APRI is one possible tool to suit this purpose because it is low cost and easy to obtain.

Little is known about the mechanisms involved in the recent emergence of severe *P. vivax* infections. Drug resistance, parasite genotype selection, and changes in the host inflammatory response may have contributed to this clinical phenomenon [40]. For this reason, the same clinical and laboratory parameters for severe *P. falciparum* infection have been used to define severe *P. vivax* malaria [41]. There is little information on inflammatory biomarkers related to severe disease. Some studies have evaluated thrombocytopenia [27], changes in platelet parameters [42], serum levels of angiopoietin I and II [15], and the presence of some non-coding RNA molecules [43] with some success but low precision. Thrombocytopenia, although frequently reported during the acute phase of vivax malaria, is not considered a predictor of severe malaria with *P. vivax* or *P. falciparum* infection [27, 28]. Similarly, other hematological

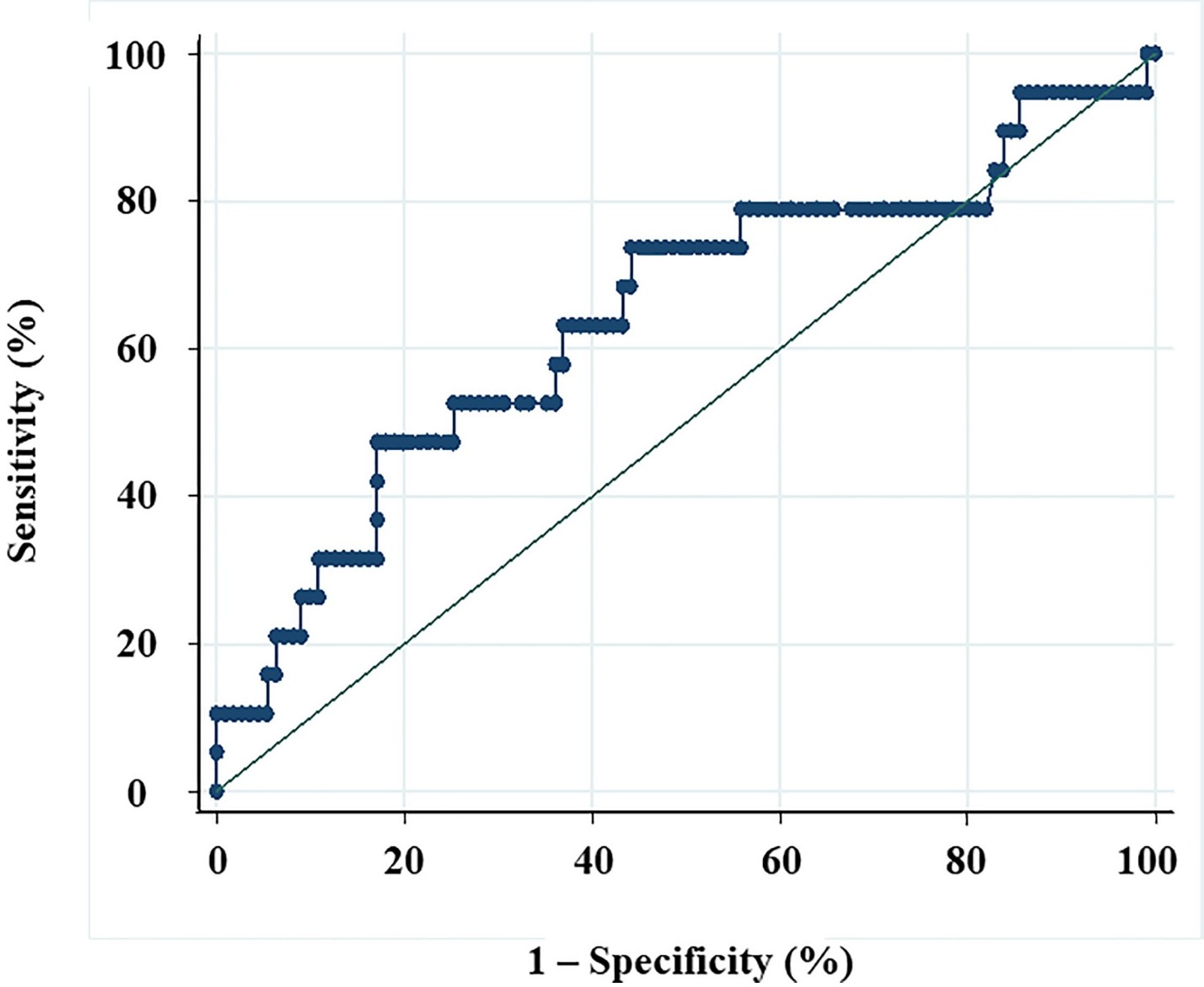

**Fig 2. ROC curve for the use of the aspartate aminotransferase-to-platelet ratio index (APRI) as a biomarker to diagnose patients at risk of severe malaria due to** *P. vivax* **infection.** A cutoff of 0.74 yielded a sensitivity of 74.0%, specificity of 56.0%, and accuracy of 65.0%.

and biochemical changes caused by infection with *P. vivax* have already been studied as diagnostic and/or prognostic markers of severe disease [27]. However, there is no information in the medical literature about the use of the APRI in malaria.

Altered hematological parameters were infrequent in the present study and were not associated with the presence of severe symptoms, except for platelet count, which was significantly lower among patients with signs of severe disease. Low platelet count associated with *P. vivax* has already been demonstrated in other studies conducted in Brazil and India [44, 45, 28, 15]. Because it occurs more frequently in *P. vivax* than *P. falciparum* infections, thrombocytopenia has already been evaluated as a parameter to distinguish between species of *Plasmodium* in human infections. However, this method was tested without success [44]. More recently, a study conducted in South America showed that thrombocytopenia was one of the most altered

hematological parameters in severe infections, regardless of species, with 90% of patients having platelet counts lower than 150,000/ μL and 43% of the population surveyed were < 50,000/ μL [46].

In the present study, serum levels of total bilirubin and liver enzymes were not different between the groups. However, it is known that *P. vivax* induces damage to the liver and can cause changes in bilirubin and hepatocytic enzyme levels, which may result in liver dysfunction [39]. A previous study showed that serum levels of bilirubin, aspartate aminotransferase, and alkaline phosphatase were significantly higher in patients considered to have severe *P. vivax* infections [12].

In recent years, the combined analysis of liver enzymes and platelet counts has provided important information regarding the role of new biomarkers in predicting neoplastic and infectious disease severity [47, 48]. One of these biomarkers, the APRI, has been widely studied as a minimally invasive marker of fibrosis and inflammation in hepatic disease [49, 50]. The APRI has also proven useful in predicting the severity of acute and systemic diseases such as dengue [19] and HELLP syndrome [51]. For this reason, we hypothesized that the APRI is altered in the acute phase of malaria since the liver is also targeted by *Plasmodium* [23].

In this study, APRI values were significantly higher among patients with danger signs of severe disease during acute *P. vivax* infection compared to those without signs of severity. This suggests a necro-inflammatory process as a pathophysiological mechanism for the increase in the APRI. The usefulness of the APRI was well-demonstrated in a 9-year longitudinal study of patients with HIV/viral hepatitis co-infection and paired controls. APRI values were stable among controls but increased 30% during the 3 years preceding death. The authors concluded that the rapid increase in APRI may predict imminent liver-related death in HIV/viral hepatitis co-infection [52]. This biomarker of hepatic fibrosis has been widely used in patients with hepatocellular carcinoma, cirrhosis, and chronic hepatitis B and C [20, 21, 53–55].

There is strong evidence of the clinical value of the APRI in predicting liver inflammation of various etiologies [56]. In a recent and pioneering study on the importance of the APRI as a predictor of signs of dengue severity, the authors concluded that the APRI proved to be an independent and more valuable predictor to discriminate patients with severe forms of the disease than individual changes in hematological and biochemical parameters [19].

In the present study, APRI accuracy for diagnosing patients at risk of severe vivax malaria was 64.5%, with sensitivity of 74.0%, and specificity of 56.0%. Despite the low specificity, these values are close to those observed in another study that used the APRI to predict chronic liver diseases in pregnant women. In the aforementioned study, the sensitivity, specificity, and accuracy were 83%, 55%, and 64%, respectively [49]. It is interesting to note that for this study, the greater sensitivity relative to specificity for the APRI does not invalidate its possible clinical application since it is preferable to identify patients with a higher risk of developing severe disease early, even at the risk of erroneous classification due to the high probability of false positives. In addition, it should be noted that the accuracy of other biomarkers proposed for the same purpose, such as the angiopoietin I/angiopoietin II ratio for the prediction of severe vivax malaria, are similar to the value reported here [15].

Some limitations should be considered in the final interpretation of the present study's results. First, there was a low number of patients with signs of severe *P. vivax* infection. Second, we adapted the criteria established by the WHO for the definition of severe falciparum malaria, mainly a parasitemia threshold of 20,000/μL, for vivax malaria.

Among the problems related to the definition of severe vivax malaria is the lack of standardized criteria for severity in both clinical and research settings. The WHO criteria have been criticized for being overly sensitive; they are clinically useful since they detect extremely severe patients, but are less useful for research because they do not identify less severe patients [29,

30]. The preference of *P. vivax* for parasitizing young red blood cells (reticulocytes) could explain the higher frequency of low parasite counts observed for this species, even in severe malaria [29, 34]. One hospital-based study conducted in Indonesia showed much lower rates of parasitism (> 6,000 parasites/μL) among hospitalized patients with severe and even fatal *P. vivax* infection [33]. The choice of a cutoff point for parasitemia of 20,000 parasites/μL was based on a study done in the Colombian Amazon, which showed parasitemia of less than 20,000 parasites/μL in 97% of patients with malaria not complicated by *P. vivax* [32].

In summary, this study indicates that routine laboratory tests, such as platelet count, liver enzyme dosage, and APRI determination, may help identify patients with greater potential for severe vivax malaria. A cutoff point of 0.74 for the APRI value has good sensitivity to identify patients with greater potential for severe disease during acute *P. vivax* infection. These findings point to the APRI as a promising biomarker of severity in acute *P. vivax* infections. However, longitudinal studies on more patients with severe disease are needed to confirm these findings and to support the clinical applicability of the APRI as a diagnostic and prognostic tool for severe acute *P. vivax* infection.

## Conclusion

Our results show that the APRI is elevated in patients with evidence of *P. vivax* infection. Based on the good APRI sensitivity found in this study, we conclude that this simple index can serve as a diagnostic biomarker to identify patients at risk of severe disease during the acute phase of *P. vivax* infection.

## Supporting information

**S1 Table. Demographic, clinical and laboratory features of the studied patients.**
(DOCX)

## Acknowledgments

We thank the medical students Andreia Nery, Natasha Crepaldi and Fabio Liberalli for their valuable support in assisting the study patients.

## Author Contributions

**Conceptualization:** Karla Sena Guedes, Bruno Antônio Marinho Sanchez, Cor Jesus Fernandes Fontes.

**Data curation:** Karla Sena Guedes.

**Formal analysis:** Karla Sena Guedes, Bruno Antônio Marinho Sanchez, Luciano Teixeira Gomes, Cor Jesus Fernandes Fontes.

**Funding acquisition:** Cor Jesus Fernandes Fontes.

**Investigation:** Cor Jesus Fernandes Fontes.

**Methodology:** Karla Sena Guedes, Luciano Teixeira Gomes, Cor Jesus Fernandes Fontes.

**Project administration:** Cor Jesus Fernandes Fontes.

**Resources:** Cor Jesus Fernandes Fontes.

**Supervision:** Cor Jesus Fernandes Fontes.

**Validation:** Karla Sena Guedes, Bruno Antônio Marinho Sanchez, Cor Jesus Fernandes Fontes.

**Visualization:** Karla Sena Guedes.

**Writing – original draft:** Karla Sena Guedes, Bruno Antônio Marinho Sanchez, Luciano Teixeira Gomes, Cor Jesus Fernandes Fontes.

**Writing – review & editing:** Bruno Antônio Marinho Sanchez, Luciano Teixeira Gomes, Cor Jesus Fernandes Fontes.

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
