## [Decision Letter · Decision Letter 0]

22 Aug 2019

PONE-D-19-21374

Aspartate aminotransferase-to-platelet ratio index (APRI): a potential marker for diagnosis in patients at risk of severe malaria caused by Plasmodium vivax

PLOS ONE

Dear Dr Fontes,

Thank you for submitting your manuscript to PLOS ONE. After careful consideration, we feel that it has merit but does not fully meet PLOS ONE’s publication criteria as it currently stands. Therefore, we invite you to submit a revised version of the manuscript that addresses the points raised during the review process.

Please notice that one reviewer actually rejected the manuscript, but after verifying the critique I understand that the issues raised may be addressed and we will proceed with the review process. 

We would appreciate receiving your revised manuscript by Oct 06 2019 11:59PM. To enhance the reproducibility of your results, we recommend that if applicable you deposit your laboratory protocols in protocols.io, where a protocol can be assigned its own identifier (DOI) such that it can be cited independently in the future. For instructions see: http://journals.plos.org/plosone/s/submission-guidelines#loc-laboratory-protocols

We look forward to receiving your revised manuscript.

Kind regards,

Leonardo Jose de Moura Carvalho

Academic Editor

PLOS ONE

Journal Requirements:

1. Please provide additional details regarding participant consent. In the ethics statement in the Methods and online submission information, please ensure that you have specified (1) whether consent was suitably informed and (2) what type you obtained (for instance, written or verbal). If your study included minors under age 18, state whether you obtained consent from parents or guardians. If the need for consent was waived by the ethics committee, please include this information

Reviewers' comments:

Reviewer's Responses to Questions

**Comments to the Author**

1. Is the manuscript technically sound, and do the data support the conclusions?

Reviewer #1: Yes

Reviewer #2: No

2. Has the statistical analysis been performed appropriately and rigorously? 

Reviewer #1: Yes

Reviewer #2: No

3. Have the authors made all data underlying the findings in their manuscript fully available?

Reviewer #1: Yes

Reviewer #2: Yes

4. Is the manuscript presented in an intelligible fashion and written in standard English?

Reviewer #1: Yes

Reviewer #2: No

5. Review Comments to the Author

Reviewer #1: The goal of the present study was to describe the clinical and laboratory characteristics of patients with acute P. vivax infection and evaluate the aspartate aminotransferase-to-platelet ratio index (APRI) as a method of identifying patients with severe disease.

Some points to be clarified:

1. What is "potentially severe disease", used several times across the manuscript. Please change to "severe disease" or "severe malaria".

2. In Introduction: "In contrast, monocyte and neutrophil counts are significantly higher in infected individuals compared to those not infected with Plasmodium [13, 14]." If I understood correctly, this sentence is related to malaria infection rather than severe malaria. Delete it.

3. What do the authors refer to as noninvasive? Techniques involving biological samples from venous puncture are not invasive? I do no agree. Change across the manuscript.

4. Methods:

"Patients with other concomitant infections, such as hepatitis B, hepatitis C, dengue, HIV, or bacterial infection, were excluded from this analysis." Were they systematically tested? Clarify.

5. "Treatment for all patients was provided in compliance with the recommendations of the Ministry of Health for the treatment of P. vivax infections [24]." Were all blood samples collected before treatment began? Clarify.

6. What is "microscopy optics"?

7. Discussion:

"Drug resistance, genomic alterations in the parasite, and changes in the host inflammatory response may have contributed to this clinical phenomenon [40]."

Genomic alterations? Change to parasite genotype selection or similar.

Reviewer #2: Aspartate aminotransferase-to-platelet ratio index (APRI): a potential marker for diagnosis in patients at risk of severe malaria caused by Plasmodium vivax

Thank you for giving me an opportunity to review this paper. I have read this manuscript with keen interest. The authors have attempted a study with two objectives; i) to describe the clinical and laboratory characteristics of patients with P. vivax infection, and ii) to evaluate the APRI as a method of identifying patients with potentially severe disease.

This is an interesting research area in the light of malaria pre-/elimination context. However, I feel that the manuscript needs substantial improvement on an array of areas.

The followings are not exhaustive, only some examples to improve the quality of presentation.

Abstract

Please, update the abstract based on the comments given below.

Line# 29…It should be better to describe as ‘the objectives’, rather than the goal.

Line #32…. I feel, this is a retrospective cohort analysis of the clinical-epidemiological data. The study data were obtained from the medical records.

Line # 38.. Please, clarify whether these are 130 patients or 130 patient’s records.

Line # 40….. It should read as ‘severe vivax malaria’.

Line # 44,45,47….. I feel, there is some exaggerations in interpretations.

The ROC value was 0.645 (0.494; 0.795), specificity was 56.0% and accuracy was 65.0%.

Line # 47…….The authors have concluded that “This simple index can serve as a diagnostic biomarker………”……….The authors need to revise the interpretation and conclusion.

TEXT

A substantial improvement in English language/grammar/ punctuation are required throughout the manuscript.

Overall, the current version of manuscript is not well written.

Background

This section is not well written. Also, it is not in sequential order. There is redundant information.

The study objectives were not well described. I feel, there are two objectives. Please, explicitly describe the objectives.

Methods

Overall, this section is not well presented.

I feel, this is a retrospective cohort analysis of the patient’s records. (Plese, see my comments to the abstract).

The authors need to improve this section. It will be better to follow the STROBE checklist.

Line# 159.. The authors assessed whether there is normal distribution of the continuous data. This is an appreciable effort.

Line # 160-61 The authors stated that they used a non-parametric Mann-Whitney test.

In Table 1 (Foot note)……. The results came out from the Fisher’s exact test/Chi-squared test for linear trend. Please, clarify.

Results

Overall, this section is not well presented.

Table 1… Title is too long.

Table 1.. Only two parameters (Platelets and APRI ) are significantly different between the two groups.

Severe TCP ( n=1), APRI > 1,5 (n = 9)

I feel, the authors can provide a correlation between these two parameters.

Line # 200……… The average APRI value was 1.24. In Table 1, it was 1.2.

Two decimal or 1 decimal?

The authors stated, “An APRI cut-off of 0.74 was reasonably sensitive and specific for identification of patients with a higher potential for severe disease”.

Please, consider that a chance of getting false positive is considerable (specificity 56%, accuracy 65%).

Line # 268-269. For this reason, it is hypothesized that the APRI is …..;; by Plasmodium [23]. This supporting information is valuable, and it will be much better to describe under “Background”.

Line # 318.. A cut-off point of 0.74 for the APRI value has good sensitivity and specificity to identify …………. I feel, an exaggeration in interpretation as 74% is not representing the specificity. please, rephrase this sentence. [Also, see my comments to the Abstract].

Thank you

6. PLOS authors have the option to publish the peer review history of their article (what does this mean?). If published, this will include your full peer review and any attached files.

Reviewer #1: No

Reviewer #2: No

---

## [Author Response · Author response to Decision Letter 0]

9 Oct 2019

Response to Reviewers 

- The authors thank the reviewers for their comments and suggestions, which they consider of great importance to improve the quality of the manuscript.

- Al lines indicated in yellow collor for each response to the Reviewer’s comment were identified in the ‘Revised Manuscript With Track Changes file.

Reviewer´s Comments to the Author

Reviewer #1: 

The goal of the present study was to describe the clinical and laboratory characteristics of patients with acute P. vivax infection and evaluate the aspartate aminotransferase-to-platelet ratio index (APRI) as a method of identifying patients with severe disease. Some points to be clarified:

1. What is "potentially severe disease", used several times across the manuscript. Please change to "severe disease" or "severe malaria".

- Since the study design did not include patient follow-up to ensure progression to severity, the authors preferred to be careful in interpreting vivax malaria severity in this cross-sectional approach only. Therefore, throughout the manuscript, the inference about severity was parsimonious. We have replaced the term "potentially serious" to "at risk of severity". [lines 32, 37, 48, 101, 104, 165, 191, 193, 195, 200, 209, 211, 2013, 217, 223, 232, 247, 249, 273-275, 289-290, 302, 304, 306, 331]. 

2. In Introduction: "In contrast, monocyte and neutrophil counts are significantly higher in infected individuals compared to those not infected with Plasmodium [13, 14]." If I understood correctly, this sentence is related to malaria infection rather than severe malaria. Delete it.

The sentence was removed as requested. [lines 69-71]

3. What do the authors refer to as noninvasive? Techniques involving biological samples from venous puncture are not invasive? I do no agree. Change across the manuscript.

- We agree with the reviewer. Blood collection is an invasive procedure, albeit a minor one. Thus we replaced the term “non-invasive” to “minimally invasive”. [lines 72, 74 e 75]. 

4. Methods:

"Patients with other concomitant infections, such as hepatitis B, hepatitis C, dengue, HIV, or bacterial infection, were excluded from this analysis." Were they systematically tested? Clarify.

- At the hospital where the study was conducted all patients undergo careful clinical examination by doctors. Epidemiological, clinical, laboratorial and demographic data of the study patients were obtained from medical records. Any evidence of other chronic infection or comorbidity in the medical records was sufficient not to include the patient in the present analysis.

- We made changes in this information in the Method section [lines 112-115, 120-122].

5. "Treatment for all patients was provided in compliance with the recommendations of the Ministry of Health for the treatment of P. vivax infections [24]." Were all blood samples collected before treatment began? Clarify.

- For all suspected malaria patients seen at the study hospital, the following laboratory tests are performed: thick smear for Plasmodium screening, blood cell count, and basic blood biochemistry. Their results are found in the patient records. 

- We clarified this in the Method section [lines 112-115, 120-122].

6. What is "microscopy optics"?

The term "microscopy optics " have been replaced in the text to "microscopic examination of the blood" [lines 125-126].

7. Discussion:

"Drug resistance, genomic alterations in the parasite, and changes in the host inflammatory response may have contributed to this clinical phenomenon [40]."

Genomic alterations? Change to parasite genotype selection or similar.

- We changed the sentence to “parasite genotype selection” as suggested. [lines 242-243]

Reviewer #2: 

Aspartate aminotransferase-to-platelet ratio index (APRI): a potential marker for diagnosis in patients at risk of severe malaria caused by Plasmodium vivax.

Thank you for giving me an opportunity to review this paper. I have read this manuscript with keen interest. The authors have attempted a study with two objectives; i) to describe the clinical and laboratory characteristics of patients with P. vivax infection, and ii) to evaluate the APRI as a method of identifying patients with potentially severe disease.

This is an interesting research area in the light of malaria pre-/elimination context. However, I feel that the manuscript needs substantial improvement on an array of areas.

The followings are not exhaustive, only some examples to improve the quality of presentation.

Abstract

Please, update the abstract based on the comments given below.

Line# 29…It should be better to describe as ‘the objectives’, rather than the goal.

- We changed the sentence to “In the present study, the aspartate aminotransferase-to-platelet ratio index (APRI) was evaluated as a method for identifying patients at risk of severe vivax malaria”. [lines 29-34].

Line #32…. I feel, this is a retrospective cohort analysis of the clinical-epidemiological data. The study data were obtained from the medical records.

- All the studied patients were outpatients and only attended the hospital for the first medical evaluation, not returning for clinical follow-up. No follow-up information on disease progression was obtained in the study. So, only a cross-sectional approach was performed. We added this information in the Methods section. [lines 116-119].

- Therefore, we only had information from the first medical evaluation. By the only cross-sectional approach we were careful not to state that the patients had severe vivax malaria. We just inferred that they were at risk for severe vivax malaria.

Line # 38.. Please, clarify whether these are 130 patients or 130 patient’s records.

- We changed to “130 patient’s records”. [line 40].

Line # 40….. It should read as ‘severe vivax malaria’.

- Since the study design did not include patient follow-up to ensure progression to severity, the authors preferred to be careful in interpreting vivax malaria severity in this cross-sectional approach only. Therefore, throughout the manuscript, the inference about severity was parsimonious. We replaced the term "potentially serious" to "at risk of severity". [lines 32, 37, 40, 48, 101, 104, 165, 191, 193, 195, 200, 209, 211, 2013, 217, 223, 232, 247, 249, 273-275, 289-290, 302, 304, 306, 331].

Line # 44,45,47….. I feel, there is some exaggerations in interpretations.

The ROC value was 0.645 (0.494; 0.795), specificity was 56.0% and accuracy was 65.0%.

- In fact, the APRI cut-off at 0.74 provided 74% sensitivity to identify patients at risk for severe vivax malaria. We highlighted the high sensitivity and the low specificity in those parts of the Discussion [line 306] and Abstract [lines 49-51].

- We highlighted the low APRI specificity in the Discussion section [lines 240-241, 336]

- As showed in lines 309-313, this find was brought to the discussion by the authors: “It is interesting to note that for this study, the greater sensitivity relative to specificity for the APRI does not invalidate its possible clinical application since it is preferable to identify patients with a higher risk of developing severe disease early, even at the risk of erroneous classification due to the high probability of false positives”.

Line # 47…….The authors have concluded that “This simple index can serve as a diagnostic biomarker………”……….The authors need to revise the interpretation and conclusion.

- We changed the study conclusion to “Our results show that the APRI is elevated in patients with evidence of P. vivax infection. Based on the good APRI sensitivity found in this study, we conclude that this simple index can serve as a diagnostic biomarker to identify patients at risk of severe disease during the acute phase of P. vivax infection.” Both in Abstract and Conclusion sections. [lines 49-51, 347-351].

TEXT

A substantial improvement in English language/grammar/ punctuation are required throughout the manuscript.

Overall, the current version of manuscript is not well written.

- The manuscript was translated and edited from Portuguese to English by EDITAGE that states that its translators have English as their native language. We will attach the Certificate of Edition along with our revised manuscript. If the revised manuscript is accepted for publication in PlosOne, we promise to resubmit it to EDITAGE for further English review.

Background

This section is not well written. Also, it is not in sequential order. There is redundant information.

- A careful review of the text presented in the Introduction (background) section has been made. Several words and phrases have been deleted, changed or inserted to make the text clearer. [lines 59, 63-64, 72-74, 75, 77-78, 83, 90, 95-97, 102-103, 106-109]

The study objectives were not well described. I feel, there are two objectives. Please, explicitly describe the objectives.

- In fact the study has only one objective. We changed the sentence to “The main objective of this study was to evaluate the APRI as a diagnostic tool for identifying patients at risk of P. vivax malaria severity.”. [lines 102-103].

Methods

Overall, this section is not well presented.

I feel, this is a retrospective cohort analysis of the patient’s records. (Please, see my comments to the abstract).

- All the studied patients were outpatients and only attended the hospital for the first medical evaluation, not returning for clinical follow-up. No follow-up information on disease progression was obtained in the study. So, only a cross-sectional approach was performed. We added this information in the Methods section. [lines 116-119].

- Therefore, we only had information from the first medical evaluation. By the only cross-sectional approach we were careful not to state that the patients had severe vivax malaria. We just inferred that they were at risk for severe vivax malaria.

The authors need to improve this section. It will be better to follow the STROBE checklist.

- According to reviewer’s suggestion we modified the methods section using the STROBE

Checklist. The STROBE checklist of all manuscript will be uploaded as a supplementary file.

Line# 159.. The authors assessed whether there is normal distribution of the continuous data. This is an appreciable effort.

- We thank the Reviewer.

Line # 160-61 The authors stated that they used a non-parametric Mann-Whitney test. In Table 1 (Foot note)……. The results came out from the Fisher’s exact test/Chi-squared test for linear trend. Please, clarify.

- In fact, the information was not complete in the Methods section and in Table 1. The four statistical tests were used: chi-square, chi-square for linear trends in proportions, Fisher’s exact test and Man-Whitney test. We changed the text of the statistical analysis in Methods, in order to complete this information. [lines 170-173, 214-215]

Results

Overall, this section is not well presented.

Table 1… Title is too long.

The table 1 title was reduced to “Characteristics of the Plasmodium vivax infected patients, according to presence and absence of signs severe vivax malaria.” [line 214]

Table 1.. Only two parameters (Platelets and APRI ) are significantly different between the two groups.

Severe TCP ( n=1), APRI > 1,5 (n = 9)

I feel, the authors can provide a correlation between these two parameters.

- This reviewer's comment was not well understood by authors. Since APRI is calculated based on platelet count, it is expected that there is a high correlation between these two parameters. The lower the platelet count the higher the APRI. 

- There is extensive information in the medical literature that platelet counts are not associated with malaria severity. This was highlighted in lines 143-146 and 259-261. Therefore, the weak correlation between AST and platelet count justifies the use of APRI as a combined measure of these two parameters for the assessment of the disease severity.

Line # 200……… The average APRI value was 1.24. In Table 1, it was 1.2.

Two decimal or 1 decimal?

- We decided to keep all APRI values to two decimals. [lines 44, 214-215, 216-218].

The authors stated, “An APRI cut-off of 0.74 was reasonably sensitive and specific for identification of patients with a higher potential for severe disease”.

Please, consider that a chance of getting false positive is considerable (specificity 56%, accuracy 65%).

- We highlighted this low specificity in the Discussion section [lines 240-241, 336]

Line # 268-269. For this reason, it is hypothesized that the APRI is …..;; by Plasmodium [23]. This supporting information is valuable, and it will be much better to describe under “Background”.

- Indeed, the intention of the authors with this sentence is to say that their initial hypothesis was that APRI would be altered in acute vivax malaria based on previous information that the liver is a important target for complications during acute malaria. Therefore, the authors prefer to keep this argument in the Discussion section. We changed the text in order to clarify this. [line 286]

Line # 318. A cut-off point of 0.74 for the APRI value has good sensitivity and specificity to identify …………. I feel, an exaggeration in interpretation as 74% is not representing the specificity. please, rephrase this sentence. [Also, see my comments to the Abstract].

- In fact, the APRI cut-off at 0.74 provided 74% sensitivity to identify patients at risk for severe vivax malaria. We highlighted the high sensitivity and the low specificity in those parts of the Discussion [line 306] and Abstract [lines 49-51].

- As showed in lines 309-313, this find was brought to the discussion by the authors: “It is interesting to note that for this study, the greater sensitivity relative to specificity for the APRI does not invalidate its possible clinical application since it is preferable to identify patients with a higher risk of developing severe disease early, even at the risk of erroneous classification due to the high probability of false positives”.

---

## [Decision Letter · Decision Letter 1]

24 Oct 2019

Aspartate aminotransferase-to-platelet ratio index (APRI): a potential marker for diagnosis in patients at risk of severe malaria caused by Plasmodium vivax

PONE-D-19-21374R1

Dear Dr. Fontes,

We are pleased to inform you that your manuscript has been judged scientifically suitable for publication and will be formally accepted for publication once it complies with all outstanding technical requirements.

With kind regards,

Leonardo Jose de Moura Carvalho

Academic Editor

PLOS ONE

Additional Editor Comments (optional):

Reviewers' comments:

Reviewer's Responses to Questions

**Comments to the Author**

1. If the authors have adequately addressed your comments raised in a previous round of review and you feel that this manuscript is now acceptable for publication, you may indicate that here to bypass the “Comments to the Author” section, enter your conflict of interest statement in the “Confidential to Editor” section, and submit your "Accept" recommendation.

Reviewer #1: All comments have been addressed

Reviewer #2: All comments have been addressed

2. Is the manuscript technically sound, and do the data support the conclusions?

Reviewer #1: Yes

Reviewer #2: Partly

3. Has the statistical analysis been performed appropriately and rigorously? 

Reviewer #1: Yes

Reviewer #2: Yes

4. Have the authors made all data underlying the findings in their manuscript fully available?

Reviewer #1: Yes

Reviewer #2: Yes

5. Is the manuscript presented in an intelligible fashion and written in standard English?

Reviewer #1: Yes

Reviewer #2: Yes

6. Review Comments to the Author

Reviewer #1: I accept as it is in this version of the manuscript "Aspartate aminotransferase-to-platelet ratio index (APRI): a potential marker for diagnosis in patients at risk of severe malaria caused by Plasmodium vivax" (PONE-D-19-21374R1). All the queries were properly responded.

Reviewer #2: The authors have fully addressed almost all comments.

For a better flow, it still need to polish English language throughout the text.

Thank you

7. PLOS authors have the option to publish the peer review history of their article (what does this mean?). If published, this will include your full peer review and any attached files.

Reviewer #1: No

Reviewer #2: Yes: Professor Cho Naing

---

## [Editor Report · Acceptance letter]

13 Nov 2019

PONE-D-19-21374R1 

Aspartate aminotransferase-to-platelet ratio index (APRI): a potential marker for diagnosis in patients at risk of severe malaria caused by Plasmodium vivax 

Dear Dr. Fontes:

I am pleased to inform you that your manuscript has been deemed suitable for publication in PLOS ONE. Congratulations! Your manuscript is now with our production department. 

With kind regards,

on behalf of

Dr. Leonardo Jose de Moura Carvalho 

Academic Editor

PLOS ONE